# Attenuated Replication of Lassa Virus Vaccine Candidate ML29 in STAT-1^-/-^ Mice

**DOI:** 10.3390/pathogens8010009

**Published:** 2019-01-15

**Authors:** Dylan M. Johnson, Jenny D. Jokinen, Igor S. Lukashevich

**Affiliations:** 1Department of Microbiology and Immunology, University of Louisville Health Sciences Center, Louisville, KY 40292, USA; 2Department of Pharmacology and Toxicology, University of Louisville Health Sciences Center, Louisville, KY 40292, USA; jenny.jokinen@louisville.edu; 3Center for Predictive Medicine for Biodefense and Emerging Infectious Diseases, NIH Regional Biocontainment Laboratory, Louisville, KY 40222, USA

**Keywords:** Lassa virus vaccine, ML29 vaccine, STAT-1^-/-^ mice, Lassa virus, Mopeia virus, interfering particles

## Abstract

Lassa virus (LASV), a highly prevalent mammalian arenavirus endemic in West Africa, can cause Lassa fever (LF), which is responsible for thousands of deaths annually. LASV is transmitted to humans from naturally infected rodents. At present, there is not an effective vaccine nor treatment. The genetic diversity of LASV is the greatest challenge for vaccine development. The reassortant ML29 carrying the L segment from the nonpathogenic Mopeia virus (MOPV) and the S segment from LASV is a vaccine candidate under current development. ML29 demonstrated complete protection in validated animal models against a Nigerian strain from clade II, which was responsible for the worst outbreak on record in 2018. This study demonstrated that ML29 was more attenuated than MOPV in STAT1^-/-^ mice, a small animal model of human LF and its sequelae. ML29 infection of these mice resulted in more than a thousand-fold reduction in viremia and viral load in tissues and strong LASV-specific adaptive T cell responses compared to MOPV-infected mice. Persistent infection of Vero cells with ML29 resulted in generation of interfering particles (IPs), which strongly interfered with the replication of LASV, MOPV and LCMV, the prototype of the Arenaviridae. ML29 IPs induced potent cell-mediated immunity and were fully attenuated in STAT1^-/-^ mice. Formulation of ML29 with IPs will improve the breadth of the host’s immune responses and further contribute to development of a pan-LASV vaccine with full coverage meeting the WHO requirements.

## 1. Introduction

Lassa virus (LASV) is a highly prevalent arenavirus in West Africa, where it infects several hundred thousand individuals annually. This results in a large number of Lassa fever (LF) cases associated with high morbidity and mortality rates [1,2]. The natural LASV reservoir is the rodent *Mastomys natalensis*, which is widely distributed throughout sub-Saharan Africa. The area where LASV is endemic covers large regions of West Africa [3], putting a population of up to 200 million people at risk for infection [4]. Furthermore, there is evidence that LASV-endemic regions are expanding [5]. The high degree of LASV genetic diversity [6,7] likely contributes to underestimates of its prevalence [8]. With the exception of dengue fever, LF has the greatest estimated global burden among all viral hemorrhagic fevers (HFs) [9]. LF outbreaks are associated with mortality rates as high as 60%, as documented during the 2015–2016 outbreak in Nigeria [10]. The currently ongoing 2018 outbreak is the worst on record [11]. There is no FDA-approved LASV vaccine or therapeutic. Treatment options are limited to off-label ribavirin with varying degrees of success. In 2017–2018, the WHO included LASV on the top priority pathogens list and issued a Target Product Profile (TPP) for LASV vaccine development [12,13,14]. 

LASV belongs to the Old World (OW) group (former lymphocytic choriomeningitis virus, LCMV-LASV sero-complex) of the genus *Mammarenavirus* of the *Arenaviridae* family [15]. The LASV genome has two RNA segments that each encode two open reading frames in opposite polarities, separated by a structured intergenic region. The L-RNA encodes for a large L protein, which functions as a RNA-dependent RNA polymerase (RdRp), and a RING finger protein Z, which comprises the matrix of virions. The S-RNA encodes for a nucleoprotein (NP) which tightly associates with viral RNA and a glycoprotein precursor (GPC) that is processed into three subunits: a stable signal peptide, SSP; a receptor-binding GP1; and a transmembrane GP2, which mediates fusion with cell membranes. All three glycoprotein subunits associate to form club-shaped features on the surface of pleomorphic viral particles, which vary in the diameter from 50 to 300 nm. In addition to LASV, the OW group includes genetically related nonpathogenic viruses: Mopeia (MOPV), Morogoro (MORV), Gairo (GAIV) and Luna (LUNV) hosted by the same rodents [16,17,18,19]. The New World (NW) mammalian arenaviruses (former Tacaribe virus, TCRV, sero-complex) comprise viruses that circulate in the Americas, and include the causative agents of South American HFs [20].

Reassortant studies using genetically related mammalian arenaviruses with different pathogenic potential demonstrated that the L gene is responsible for high levels of virus replication in vivo, and is associated with acute disease in experimental animals [21,22]. Coinfection of cells with LASV and MOPV resulted in generation of MOPV/LASV reassortants [23,24]. One of the rationally selected clones, ML29, carrying the L-RNA from MOPV and the S-RNA from the Josiah strain of LASV (LASV/JOS), is a promising LASV vaccine candidate [24,25,26]. 

While MOPV L-RNA is the major driving force of ML29 attenuation, 18 mutations incorporated in the ML29 genome during in vitro selection additionally seem to contribute to the attenuation of ML29 [25,27]. Indeed, transcriptome profiling of human peripheral blood mononuclear cells (hPBMC) exposed to LASV or ML29 exhibited distinct molecular signatures that can be useful biomarkers of pathogenicity and protection [28,29]. Similar studies comparing MOPV and ML29 infected hPBMC revealed that gene expression patterns in mock and ML29 infected hPBMC clustered together and differed from the expression pattern observed in MOPV infected hPBMC [28]. 

Recently, recombinant ML29 (rML29) was rescued from cDNA clones and demonstrated the same features as a biological ML29 (bML29) [30]. Reverse genetics provides a powerful tool for elucidation of the individual contributions of ML29-specific mutations to attenuation. However, the absence of a reliable small animal model for LASV pathogenicity is a major obstacle for the field. LASV, as well as other mammalian arenaviruses, stimulate antiviral immunity differently in natural rodent hosts and in humans and non-human primates, NHPs [31,32,33]. While some pathogenic features of human LF can be observed in strain 13 guinea pigs [34,35,36], there is a poor correlation between clinical outcome of LF in humans and virulence of LASV in guinea pigs [37]. Results of LASV vaccination/challenge studies in strain 13 guinea pigs are not reproducible in NHPs [32,38,39]. In addition, strain 13 guinea pigs are no longer commercially available and difficult to breed. 

Rodent models remain useful to study some mechanisms of LF pathogenesis and protective immune responses, and to provide valuable information during preclinical development of vaccine candidates [31,40]. Previous research has documented that mice lacking a functional STAT1 pathway are highly susceptible to LASV infection, and develop fatal disease with some pathological features that mimic human LF [41]. Notably, human LASV isolates from fatal LF cases in Sierra Leone induced clinically similar fatal disease including high viral load in blood and visceral organs of STAT-1^-/-^ mice after intraperitoneal inoculation (i.p.). In contrast, LASV isolated from a non-lethal human case induced disease with moderate mortality. Surviving mice developed hearing loss that mimicked the sensorineural hearing loss (SNHL) commonly observed in LF patients during convalescence [42].

In this study, we provided additional evidence of deep ML29 attenuation by modeling the LASV infection protocol for STAT^-/-^ mice as a means to assess the safety profile of ML29 (bML29 or rML29) in comparison with MOPV. We also generated ML29 stocks enriched with interfering particles (IPs) produced by ML29-persisitently infected cells. The growing body of evidence indicates that, during natural infection, IPs are critically involved in the modulation of viral load, innate immune responses, disease outcome, and contribute to the evolutionary persistence of viruses [43,44,45]. An accumulation of IPs with genome deletions has been documented during the production of live-attenuated polio [46], influenza [47,48] and measles [49,50,51] vaccines, and was associated with modulating vaccine potency and immunostimulatory properties. Here, we demonstrated that STAT^-/-^ mice are a susceptible model capable of discriminating between viruses with different levels of attenuation, MOPV and ML29. ML29 IPs were completely attenuated in these mice, enhancing immunogenic features of ML29. The broad cross-protection activity of ML29 IPs and their intrinsic adjuvant features can be potentially used for rational formulation of a pan-LASV ML29-based vaccine by developing vaccine-manufacturing protocols to optimize the ratio between infectious ML29 and IPs. 

## 2. Results

### 2.1. Generation of Interfering Particles (IPs) by ML29-Persisitently Infected Cells

Arenavirus IPs can be generated at high multiplicity of infection (MOI) by serial infection of fresh tissue cultures with undiluted culture medium harvested from previous passages. Alternatively, initially infected cells can be subjected to serial passages. Both methods resulted in the accumulation of IPs. Earlier studies suggested a contribution of IPs in the establishment and/or maintenance of LCMV persistent infection in vitro and in vivo, and in the inhibition of virus-induced immunopathology in mice [52,53,54,55,56,57,58]. The protocol previously applied for the generation of LASV IPs in cell cultures was used to establish ML29 persistent infection of Vero cells. In line with previous observations [59,60,61], passages of ML29-infected cells, either infected with bML29 or with rML29, resulted in the gradual decline of infectious particles released in culture medium. As seen in Figure 1, the infectious titers of ML29 released into the medium of persistently infected Vero cells (Vero/ML29) were reduced almost 100-fold after 10 passages, and was under detectable levels after 15 passages. Vero/ML29 cells strongly interfered with replication of homologous (ML29) or genetically related LASV/JOS, LCMV, and MOPV and did not generate infectious plaques after super infection of Vero/ML29 with these viruses. Meanwhile, neither the replication of TCRV, a NW arenavirus, nor the nonrelated Ebola virus, was affected in these cells. ML29 IPs released from persistently infected Vero/ML29 cells were able to suppress replication of homologous virus in a dose-dependent manner (Figure 1f).

While the replication of ML29 or LASV was not detectable in ML29/Vero cells after passage 10, viral proteins were detected in these cells by immunofocus (IF) assay (Figure 1c). In contrast to infectious “plaques”, this assay detects cells expressing virus antigens stained by specific antibodies. With an increasing number of passages, ML29-infected cells lost their ability to produce plaques after homologous superinfection, but were strongly positive for ML29-specific antigens as demonstrated by quantitated analyses (Figure 1c; Appendix A, Figure A1). Differences in the morphology of IF foci between naïve and Vero/ML29 cells after infection and superinfection with ML29, respectively, seems to be related to differences in the expression of NP and GP proteins in different passages of Vero/ML29 (Appendix A, Figure A1F).

The viral RNA load in Vero/ML29 cells, as assessed by quantitative qRT-PCR targeting the NP gene, declined about tenfold during the first five passages and persisted approximately at the same levels during subsequent passages of Vero/ML29 (Figure 1b). The viral RNA load remained at a similar level through 50 passages (data not shown). Our attempts failed to detect genomic RNA deletions using RT-PCR, either in infected cells or in virions released from infected cells (not shown, see Discussion).

### 2.2. MOPV and Attenuated Reassortant ML29 Induce Experimental Disease with Different Clinical Manifestations and Outcome in STAT-1^-/-^ Mice

In STAT1^-/-^ mice, LASV infection resulted in experimental disease with the outcome depending on the pathogenic potential of the LASV isolate [42]. Based on these observations, we tested the attenuation of MOPV and ML29 in these mice. Using the LASV infection protocol, three groups of mice (n = 9) were IP inoculated with MOPV, ML29 and ML29P50 (IPs produced by Vero/ML29, passage 50). MOPV infection resulted in poor grooming, lethargy and signs of dehydration starting around day 8-post-infection. These symptoms aggravated, and the mice gradually lost weight, and experienced a drop in body temperature. All animals in this group met euthanasia criteria within 11–17 days following infection (Figure 2). In ML29 infected STAT1^-/-^ mice, the symptoms started to improve at the late stage of the infection. Body temperature quickly recovered after day 10 and 33% of mice survived at day 21. No clinical signs were observed in the ML29P50 group. These mice gained weight at the end of the observation period and their temperature fluctuated within a normal range. 

As a control, wild-type mice with an identical genetic background were i.p. inoculated with MOPV, ML29 and ML29P50. The mice were successfully infected with these viruses as was documented by detection of viral RNA in tested tissues (Appendix A, Figure A2). However, in the wild-type mice, the replication of the viruses was well-controlled and viral RNA was barely detectable at day 21. As expected, infection did not induce clinical manifestations in any of the wild-type study groups, and these mice gained weight at the end of the 21-day observation period.

### 2.3. Replication of MOPV and ML29 in Tissues of STAT-^-/-^ Mice: No Viremia and Infectious Virus in Tissues of ML29P50-Infected Mice

High viremia, infectious titers, and viral RNA loads were associated with replication of MOPV in tissues of STAT-1^-/-^ mice as assessed by infectious plaque assay and qRT-PCR (Figure 2). At the peak of clinical manifestation, days 7–10, viremia and viral load in the liver reached more than 1 × 10^7^ PFU/mL or PFU/g, respectively. RNAscope in situ hybridization confirmed extensive liver infection with strong signals associated with hepatocytes and endothelial cells (Figure 3, arrowed). In contrast, replication of ML29 was more attenuated in STAT-1^-/-^ mice. Viremia and viral load in the liver was 3-logs lower in these mice compared with MOPV infection, and only a few weakly positive hepatocytes were seen in ML29-infected liver samples. In brain tissue, replication of infectious ML29 was transiently detected at early and late stages of infection. Replication of infectious ML29P50 in STAT-1^-/-^ mice was below detectable levels in blood, liver and brain. However, replication of ML29P50 was detected in tissues by viral RNA-based assays, qRT-PCR, and RNAscope in situ hybridization. As seen in Figure 2g–i, at early stage of the infection, relative levels of ML29P50 viral RNA in tissues were lower than MOPV and ML29 viral RNAs. ML29P50 was barely detectable in the tissues at late stage of the infection. This indicates that ML29P50 was well-controlled, an observation that is in line with the absence of clinical manifestations.

### 2.4. Host Responses in Mice Infected with MOPV and ML29

Cell-mediated immune responses in infected mice were assessed by ELISPOT using stimulation of spleen cells isolated on day 3, 7, and 21 after infection with peptide cocktails derived from LASV/JOS GPC and MOPV GPC as previously described [62,63]. At day 3, virus-specific T cells secreting IFN-ɣ, IL-2 or both were barely detectable in all groups of mice (not shown). By day 7, in STAT-1^-/-^ mice, comparable levels of IFN-ɣ-secreting cells (~0.4% of total spleen cells) were observed in spleens from ML29- and ML29P50-infected mice (Figure 4a). In wild-type mice, ML29 and ML29P50 infection induced slightly lower T-cell responses Notably, in wild-type and STAT-1^-/-^ mice infected either with ML29 or with ML29P50, similar levels of IFN-ɣ-secreting cells were induced after stimulation with GPC cocktails derived from LASV or MOPV. Infection with MOPV induced comparatively diminished T cell responses in wild-type and in STAT-1^-/-^ mice, predominantly after stimulation with homologous MOPV GPC-derived peptides. After stimulation with GPC cocktails, spleen cells isolated from mice infected with ML29 and ML29P50, but not with MOPV, also secreted IL-2 cytokine, ~0.1% of total spleen cells. Cells secreting both cytokines were easily detectable at day 7 in ML29 and ML29P50 mice after stimulation with homologous antigens. In MOPV infected STAT-1^-/-^ mice, this cells population was barely detectable. 

Survival of ML29P50-infected mice was clearly associated with strong cell-mediated responses. As seen in Figure 4, in spleens isolated from STAT-1^-/-^ mice on day 21, cells secreting IFN-ɣ, IL2, or both were strongly upregulated after stimulation with GPC cocktails, >0.3–0.6%. In wild-type mice inoculated with ML29P50, these responses had declined by this time point. Interestingly enough, in both groups of mice infected with ML29P50 humoral responses were similar (Figure 4g).

Measurement of pro-inflammatory cytokine expression in spleen of STAT-1^-/-^ mice infected with MOPV revealed upregulation of IL-6 and IL-1β at early stages of the infection and rapidly declined at later time points while TNF-α was downregulated for all viruses except the early stages of MOPV and the late stage of ML29P50 (Figure 5, Appendix A, Figure A3). Levels of IL-1β and IL-6 mRNA in spleens of the mice infected with ML29 and ML25P50 were upregulated at day 3 and day 7, respectively. At the later time point, the infection had minimal if any effect on TNF-α and IL-1β expression in STAT-1^-/-^ and control mice. In MOPV-infected wild-type mice, IL-6 mRNA levels had similar kinetics with those in STAT-1^-/-^ mice, with downregulation in the liver and spleen tissues during the infection.

## 3. Discussion

The first aim of this study was to determine if STAT-1^-/-^ mice provide a reliable small animal model to assess level of attenuation of LASV-related arenaviruses. LASV and MOPV share a common reservoir, *M. natalensis*, found in the western and southeastern regions of Africa [16,17,64]. In Mozambique and Tanzania, areas where MOPV and MORV are prevalent, there is a marked absence of clinical cases of LF. Additionally, MOPV does not cause clinical signs in guinea pigs and NHPs [35]. These observations, combined with ability of MOPV to generate in vitro replication-competent reassortants with LASV [24], and to protect experimentally infected NHPs against fatal LF [65], suggests that MOPV is a “naturally attenuated” genetic relative of LASV.

Studies to assess virulence of human LASV isolates in NHPs are very limited [66]. LASV isolates from LF patients with different clinical outcome showed a weak correlation between severity of human disease and virulence in guinea pigs [35,37]. Meanwhile, in STAT-1^-/-^ mice, outcome of the disease correlated with virulence of human LASV isolates. Four out of 5 mice infected with a lethal isolate met euthanasia criteria between days 7 and 8, while mice infected with nonlethal LASV isolates developed chronic disease [41,42]. In our experiments, MOPV infection of STAT-1^-/-^ mice induced manifested disease that met euthanasia criteria 11–17 days post-infection (Figure 2a). The delayed euthanasia time-point of 11–17 days versus 7–8 days indicates some level of attenuation of MOPV infection in STAT-1^-/-^ mice. However, high levels of viremia and viral load in target tissues, comparable with those in animals infected with virulent LASV isolates [41,42], and poor adaptive immune responses (Figure 4) resulted in aggravated disease. In contrast to MOPV, ML29 infection in STAT-1^-/-^ mice was more attenuated, and 33% of animals survived in this group (Figure 2a). Compared to MOPV-infected mice, ML29 infection notably resulted in more than a thousand-fold reduction in viremia and viral load in tissues by plaque assay (Figure 2d–f). Strong T cell responses contributed to viral control in this experimental group (Figure 4). It is well documented that strong T cell-mediated immunity correlates with protection and positive clinical outcomes in experimentally infected NHPs and LF patients [33,67,68,69].

The second aim of this study was to assess the contribution of IPs generated by ML29-persistently infected cells to the attenuation and modulation of immune responses. Similar to many other RNA viruses, mammalian arenaviruses can generate IPs during acute infection of cells at high MOI, or during persistent infection. The defective IPs: (i) are antigenically identical to parental viruses and contain the same structural proteins; (ii) preserve 5′ and 3′ terminal sequences of parental genome but have extensive internal deletions; (iii) can only replicate with help of standard virus; and (iv) strongly compete for replication machinery with the parental virus [70,71,72,73,74]. Historically, arenavirus IPs were observed following in vitro infection at high MOI, and masked the cell-killing potential of standard viruses. Earlier studies suggested the contribution of IPs in the establishment and/or maintenance of LCMV persistent infection in vitro, and in inhibition of virus-induced immunopathology in mice [52,53,54,55,56,57,58]. Similarly, LASV IPs generated by persistently infected cells strongly interfered with replication of standard LASV in vitro, were attenuated in C3H mice, and partially protected mice against wild-type LASV challenge [59,60,75,76]. In contrast to IPs generated from VSV, Sendai, Sindbis, and other RNA viruses, arenavirus IPs are difficult to separate from standard virions [55]. The unique ability of mammalian arenaviruses to incorporate host ribosomes and RNAs during the late stage of virus maturation [15] seems to be responsible for failure of density-based methods to purify arenavirus IPs. Nevertheless, LCMV defective IPs partially purified from culture medium of persistently infected cells lacked the S-RNA segment [77], compared to LASV defective IPs where the L RNA segment was not detectable [61]. In line with these results, UV inactivation experiments confirmed “smaller” genomes for LCMV [54] and LASV defective IPs [61]. The LCMV L-RNA segment was much less abundant and not detectable during the early stage of persistent infection. However, RNA deletions were not found among virus-specific RNA species in brain tissue of LCMV persistently infected mice. During the progression of persistence, an accumulation of terminally truncated RNA species was documented [78].

In our experiments, we were able to demonstrate the basic features of ML29 IPs generated by persistently infected cells (gradual infectivity loss during passages, antigenic identity and persistence, GP/NP ratio fluctuation, interference with homologous and closely related viruses) (Appendix A, Figure A1). However, we failed to detect genomic deletions among virus-specific RNA species extracted from persistently infected cells or among viral RNA extracted from concentrated supernatants by RT-PCR methods using different sets of primers to amplify the genomic L and S RNA segments. By using qRT-PCR quantification of the LASV(ML29) NP gene, we demonstrated that S-RNA copies rapidly declined during the first five cell culture passages of infected cells and the number of copies persisted at roughly same levels during passages 10–25 (Figure 1b). These levels remain relatively constant for all subsequent passages tested through passage 50 (not shown). In contrast to the permanent level of the S-RNA replication and protein expression detected by IF assay (Figure 1d), the infectivity of particles produced by persistently infected cells dropped dramatically during the first 10 passages, was below the threshold of detection by passage 15 (Figure 1a), and was not detectable at the final passage 50. 

The inability of ML29-persistently infected cells to generate infectious plaques after homologous superinfection with ML29, or infection with LASV/JOS or MOPV, can be partially explained by arenavirus Z-mediated “superinfection exclusion” [79]. However, the replication of the NW mammalian arenavirus TCRV was not affected in ML29/Vero cells (Figure 1f). Recent study documented the capability of LCMV, LASV and MACV mammalian arenaviruses to replicate and disseminate without the Z protein providing evidence that NP can play role as potential surrogate of the Z protein [80]. We assume that ML29 IPs-induced interference was unlikely to be related to Z protein. Nevertheless, the contribution of the Z protein to innate immune responses cannot be fully excluded. ML29 has Z protein derived from MOPV. Interestingly enough, Z protein of pathogenic arenaviruses (including LASV,) but not that of nonpathogenic viruses, including the MOPV Z, inhibit RIG-I-like receptor-dependent IFN type I production [81].

In this study, we failed to provide evidence that ML29 IPs produced by ML29/Vero cells are “classical” defective IPs and the nature of ML29 IP-based interference has to be further elucidated. Nevertheless, ML29 IPs seem to be very attractive for vaccine formulation and development. First, as seen in Figure 2a, ML29P50 generated by ML29/Vero cells, passage 50, were completely attenuated in STAT-1^-/-^ mice. These particles were not detectable by plaque assay (test on infectivity) in blood and tissues of mice. However, the level of ML29P50 RNA replication was similar or only slightly reduced in comparison with “wild-type” ML29 RNA replication (Figure 2g–i). Second, while ML29P50 and ML29 induced comparable T cell responses at day 7, at the late stage of the infection ML29P50 generated much stronger T cell immunity as assessed by IFN-ɣ/IL-2 ELISPOT in line with the strong immunomodulation potency of defective IPs [82]. While the mechanism of LASV protection can be dependent on the vaccine platform itself, as well as the animal challenge protocol—in the case of the ML29 vaccine, T cell immunity, but not antibody production—was associated with protection of NHPs [83]. Third, ML29 is the only vaccine with documented evidence of T-cell-mediated cross-protection against Nigerian LASV strains from clade II, among currently available LASV vaccine candidates [84]. This is critical considering the Nigerian LASV strains from clade II are responsible for causing the current unprecedented LF outbreak. LASV genetic diversity is the greatest challenge for vaccine development. The formulation of ML29 with IPs will improve the breadth of the host’s immune responses [85] and further contribute to the development of a pan-LASV vaccine with full coverage meeting the WHO requirements [13].

## 4. Materials and Methods

### 4.1. Viruses and Cells

Generation of MOPV An20410, LASV/Josiah, and MOP/LAS reassortant (ML29) LCMV (WE and ARM) stocks in Vero cells and plaque titration technique was previously described [25]. Freeze-dried Tacaribe virus (TCRV-11573, ATCC^®^ VR-1272CAF™) was purchased from ATCC and the virus stock was prepared by low MOI passage in Vero cells. Infectious plaques for all viruses, except MOPV, were counted at 5 days after infection by treatment of infected cell monolayers under agarose overlay with 4% paraformaldehyde and staining with 1% crystal violet. For MOPV, neutral red stain staining was used for plaque development. Vero C1008 cells (Vero 76, clone E6, ATCC^®^ CRL1586) were maintained in DMEM/F-12 supplemented with 10% fetal calf serum, 1X antibiotic-antimycotic (ThermoFisher), and 1X Glutamax (ThermoFisher). ML29 persistent infection in Vero cells was established by using a previously described protocol [59,61]. In brief, after the initial infection of Vero cell monolayers with an MOI of 1.0, at weekly intervals, cells were detached from cell tissue flasks by treatment with 0.05% trypsin-EDTA (ThermoFisher) and subcultured at a 1:10 ratio. Culture medium was changed at 3–4-day intervals. Samples of cells and/or culture medium were used for plaque titration, antigen detection assays (IF, Western) and for RNA isolation. To titrate ML29 IPs produced by persistently infected Vero/ML29 cells, monolayers of naïve Vero cells were pretreated with dilutions of Vero/ML29 passage 5, 20 and 30 supernatants for 1 h followed by infection with ML29 and detection of plaques as described above. LASV/JOS or EBOV plaque titrations were performed in BSL-4 facilities using maximum containment practices at the University of Texas Medical Branch with the assistance of Dr. Slobodan Paessler’s lab. 

### 4.2. Immunofocus Assay

Vero E6 cells or ML29 persistently infected cells (Vero/ML29) were plated at a density of 2 × 10^4^ cells/well in 96-well tissue culture plates. Twenty-four hours later, cells were overlaid with methylcellulose media. Naïve or Vero/ML29 cells were infected with 0.1 MOI of ML29 for 1 h prior to being overlaid with methylcellulose. After 3 days, the methylcellulose was removed and cells were then fixed with an acetone methanol mixture. Residual endogenous peroxidase was blocked by hydrogen peroxide and cells were incubated overnight in 10% FBS. After washing, cells were treated with either polyclonal monkey-anti-ML29 antibody (1:400), monoclonal mouse-anti-LASV/JOS-NP, 1:100 (GenScript, Piscataway, NJ, USA) or polyclonal rabbit-anti-LASV-GP 1:400 (IBT Bioservices, Rockville, MD, USA) followed by HRP linked appropriate secondary antibody. TrueBlue Peroxidase Substrate (SeraCare, Milford, MA, USA) was used for detection. Quantification of staining from high-quality images of scanned plates was performed using Adobe Photoshop to reduce background and threshold images followed by ImageJ counting of pixels. 

### 4.3. Animal Protocols

Four-to-five week-old female STAT-1^-/-^ mice (129S6/SvEv-Stat1tm1Rds) and wild-type controls (129SVE-F) were purchased from Taconic (Hudson, NY, USA). During a 1-week acclimation, a temperature and identification transponder was implanted subcutaneously and the mice were transferred to ABSL-3 housing at the NIH Regional Biocontainment Laboratory on the University of Louisville campus. For infection, 1 × 10^3^ PFU of MOPV, ML29 or ML29P50 (quantitated as a qRT-PCR equivalent dose) was administered (i.p.) in 100 μL of PBS. Mice were monitored daily during 21 days and any animal with 25% weight loss was determined to have met the humane euthanasia criteria. Plasma samples from infected mice that had been euthanized were collected in EDTA tubes (BD). Tissue sample homogenates (10% *w*/*v*) were prepared with an Omni TH Tissue Homogenizer in DMEM/F12 followed with centrifugation at 4500× *g* for 20 min. Clarified tissue homogenates were tested in the plaque assay described above. All animal protocols were approved by the University of Louisville Institutional Animal Care and Use Committee.

### 4.4. qRT-PCR and RNAscope In Situ Hybridization

Tissue samples were placed into 2-mL screw-top tubes with 1 mL TRIzol Reagent (ThermoFisher) and glass beads, processed in a bead homogenizer, and tissue homogenates were stored at −80 °C until RNA isolation by phenol-chloroform extraction and ethanol precipitation. The qScript cDNA super mix (Quanta Biosciences) was used to make cDNA with an input of 1000 ng of RNA. qRT-PCR with TaqMan primers targeting the MOPV L gene, the LASV NP gene, IL-6, TNF-α, or IL-1β were used with a 18S housekeeping probe on a StepOnePlus qRT-PCR analyzer (Applied Biosystems). All IL-6, TNF-α, IL-1β and 18S primer/probe sets were commercially purchased through ThermoFisher Scientific through the Applied Biosystems division. Primer/probe set for MOPV amplified a region of the L segment (FWD 5′-TCCTCAATTAGG CGTGTGAA-3′; REV 5′-TACACATCCTTGGGTCCTGA-3′; probe 6FAM-CCCTGTTCCCTCCAACTTGTTCTT TG-TAMRA). Primer/probe set for LASV (ML29) amplified the NP gene segment (FWD 5′-TCC AACATATTGCCACCATC-3′; REV 5′- GCT GAC TCA AAG TCA TCC CA-3′; probe 6FAM TGCCTTCACAGCTGCACCCA-TAMRA). The qRT-PCR reaction contained 5 μM of each primer and 2 μM of probe for each primer/probe set, 5 µL of cDNA and TaqMan Fast Advanced Master Mix (ThermoFisher) in a final reaction volume of 20 μL. The reaction conditions were as follows: 50 °C for 2 min, 95 °C for 20 s then 40 cycles alternating between 95 °C for 3 s and 60 °C for 30 s. For RNAscope hybridization, fixed tissue sections were embedding in paraffin, cut on a Leica RM2125 RTS microtome, and mounted on glass slides. Custom designed target probes, preamplifier, amplifier, and label probe targeting MOPV L RNA were synthesized by Adanced Cell Diagnostics (Hayward, CA, USA) and RNAscope assay was performed according to the manufacturer’s manual. Chromogenic detection was performed using DAB followed by counterstaining with hematoxylin.

### 4.5. EISPOT and IgG ELISA

Assessment of T cell responses in infected mice by the murine IFN-γ/IL-2 Double-Color Enzymatic ELISPOT (Cellular Technology Ltd., Cleveland, OH, USA) has been recently described [63]. In brief, erythrocyte-free splenocytes were added to 96-well filter plates (Millipore, MSIPS4510) pre-coated with anti-mouse cytokine antibody in triplicate at a density of 4 × 10^5^ cells/well. Cells were stimulated overnight at 37 °C with cocktails of 10 µM LASV/JOS or MOPV GPC peptides (Mimotopes Ltd, Melbourne, Australia). Each cocktail contained 69 overlapping 20-mer peptides. As positive and negative controls, Conconavalin A (ThermoFisher) and CLT media alone was added to quality control wells. After stimulation, plates were developed according to the manufacturer’s protocol and cells secreting individual cytokines IFN-γ or IL-2 or both were counted using C.T.L. Ltd. Immunospot^®^ S5 Micro-analyzer and Immunospot^®^ V 4.0 software. Quality control analysis was provided by C.T.L. Ltd. 

IgG ELISA was performed as previously described [62]. In brief, microtiter plates were coated with 5 × 10^6^ PFU/well of sonicated virus suspension in carbonate buffer (Sigma) overnight. Plates were washed, blocked with 10% milk for 2 h, washed again, and coated with dilutions of mouse serum for 1 h. Plates were washed and secondary HRP linked rabbit-anti-mouse IgG (Sigma) was added at a 1:2500 dilution for 1 h. Plates were washed and SureBlue TMB peroxidase substrate (SeraCare) was added for 15 min before the addition of TMB Stop Solution (SeraCare).

### 4.6. Statistics Analysis

Results are reported as means ± SEM (n = 4–5). ANOVA with Bonferroni’s post-hoc test (for parametric data) or the Mann–Whitney rank-sum test (for nonparametric data) was used for the determination of statistical significance among treatment groups, as appropriate. Statistical analysis (mean, SD, T-test) and graphics were performed using the GraphPad Prism version 7 for Windows package (GraphPad Software, LaJolla, CA, USA).

## Figures and Tables

**Figure 1 pathogens-08-00009-f001:**
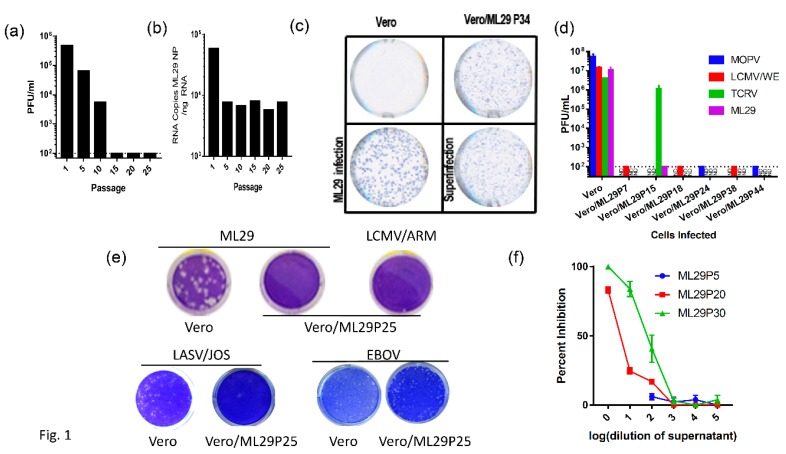
Vero cells persistently infected with ML29 interfere with replication of homologous and genetically related arenaviruses. (**a**) Supernatants from a Vero cell line persistently infected with ML29 (Vero/ML29) were titrated for infectious virus by plaque assay at the indicated passages. (**b**) Total RNA was isolated from a Vero/ML29 cells at the indicated passages and was assayed using qRT-PCR with a LASV/JOS nucleoprotein (NP) TaqMan probe. Copy number was determined by regression against a standard curve. (**c**) Immunofocus assay using polyclonal moneky-anti-ML29 on uninfected Vero E6 cells (top left), Vero/ML29 passage 34 cells (top right), Vero E6 cells infected with a 0.1 MOI of ML29 for 3 days (bottom left), and Vero/ML29 passage 34 cells superinfected with a 0.1 MOI of ML29 for 3 days (bottom right). (**d**) Naïve or persistently infected Vero/ML29 cells at different passages were used to titer MOPV, LCMV/WE, TCRV, ML29 by plaque assay. (**e**) Plaque-forming activity of Vero/ML29 cells. ML29 (top left and top center), LCMV/ARM (top right), LASV/JOS (bottom left and bottom left center), and EBOV (bottom right center and bottom right) were used in a plaque assay on Vero E6 cell (top left, bottom left, and bottom right center) or the 25th passage of a Vero cell line persistently infected with ML29, Vero/ML29P25 (top center, top right, bottom center left, bottom right.) (**f**) Dose-dependent effects of IPs on replication of ML29. Vero E6 cells were pre-incubated for 1 h with the indicated dilution of ML29 IPs produced by Vero/ML29 cells at passage 5, 20 or 30. The pretreated cells were then infected with standard ML29 and viral titer was determined by plaque assay. Percent inhibition is normalized to infectious titer determined on Vero E6 cells that were not pretreated.

**Figure 2 pathogens-08-00009-f002:**
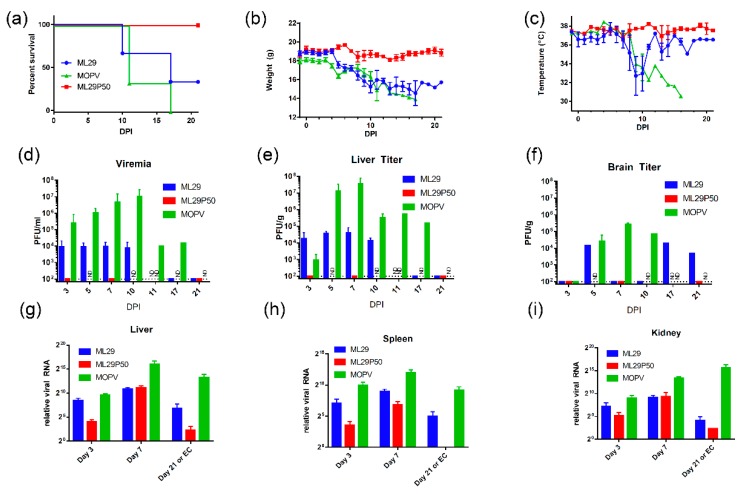
Clinical manifestation and outcome of MOPV and ML29 infections in STAT-1^-/-^ mice. (**a**) Mice were infected (i.p.) with 1 × 10^3^ PFU of MOPV An20410, ML29 (standard infectious virus) or with ML29P50 (IPs collected from culture medium of Vero/ML29P50 cells). ML29P50 IPs were quantified by MOPV L gene qRT-PCR and used in genome-equivalent dose, 1 × 10^3^ PFU. (**b**) Weight and (**c**) Temperature of infected mice. (**d**) Viremia, (**e**) Infectious viral load in liver, and (**f**) brain of STAT-1^-/-^ mice. (**g**–**i**) Viral RNA load assessed as relative expression of MOPV-L gene by qRT-PCR in the indicated tissues at the time point indicated. DPI, days post-infection. EC, endpoint criteria.

**Figure 3 pathogens-08-00009-f003:**
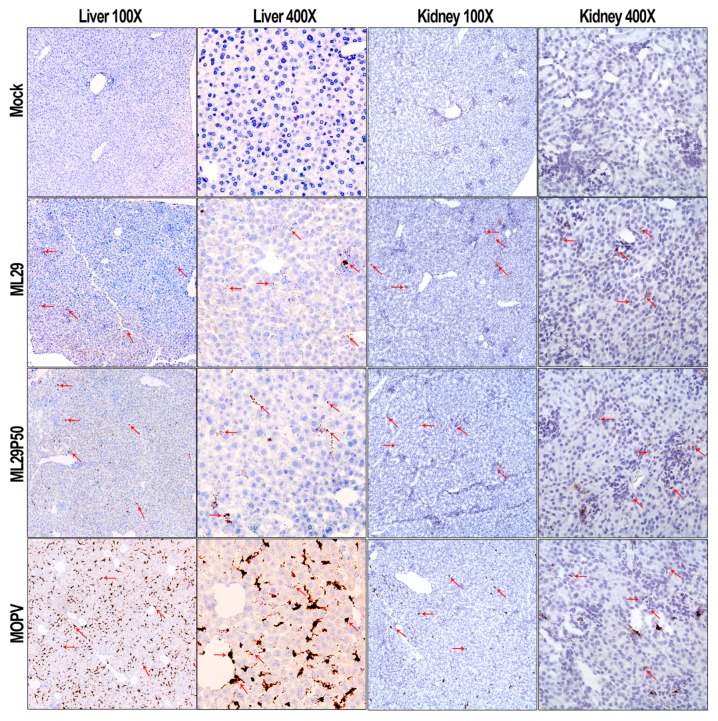
RNAscope in situ hybridization. Representative images at two different magnifications (100× and 400×) of RNAscope in situ hybridization (brown spots, arrowed) targeting the MOPV-L gene in the liver and kidney of STAT1^-/-^ mice.

**Figure 4 pathogens-08-00009-f004:**
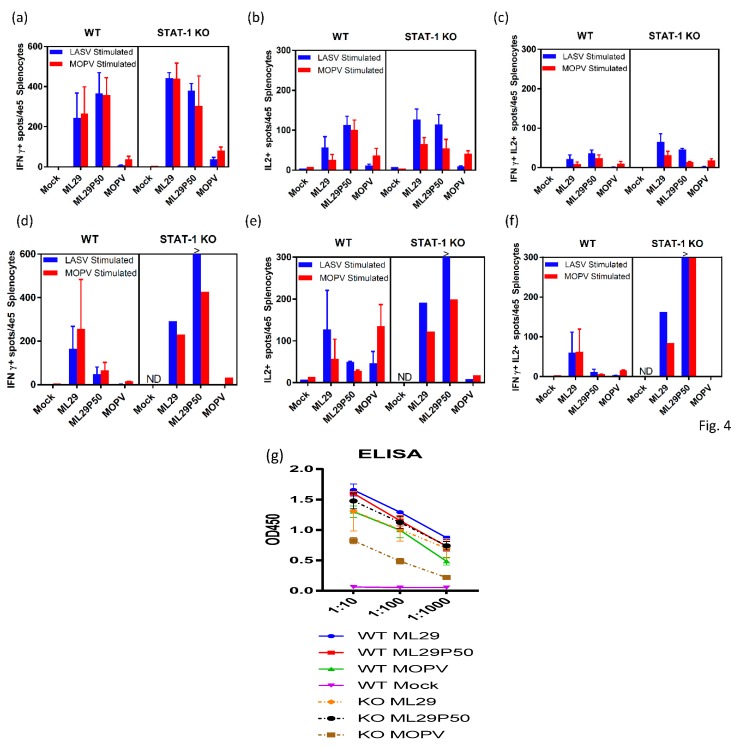
Adaptive immune responses of STAT-1^-/-^ mice. ELISPOT counts per 4 × 10^5^ splenocytes from STAT-1^-/-^ mice or wild-type (WT) controls collected (**a**–**c**) 7 days post-infection or (**d**–**f**) at endpoint criteria or 21 days post-infection. Splenocytes were stimulated with LASV GPC or MOPV GPC-derived peptide cocktails for 24 h. (**a**,**d**) IFN-γ-secreted cells, (**b**,**e**) IL-2-secreted cells, or (**c**,**d**) poly-functional IFN-γ/IL-2-secreting cells. (**g**) Antibody responses measured by IgG ELISA in the serum of STAT-1^-/-^ mice or wild-type controls collected at 21 days post-infection or endpoint criteria. >, Too numerous to count.

**Figure 5 pathogens-08-00009-f005:**
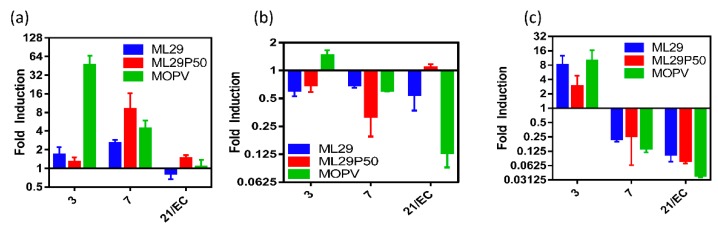
Cytokine responses of STAT-1^-/-^ mice. (**a**) IL-6, (**b**) TNF-α, or (**c**) IL-1β expression in splenocytes of STAT-1^-/-^ mice relative to mock infected by qRT-PCR.

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
