# Peer review of "Attenuated Replication of Lassa Virus Vaccine Candidate ML29 in STAT-1-/- Mice"

_pathogens, 2019, doi:10.3390/pathogens8010009_

Round 1

Reviewer 1 Report

ML29 is a LASV/MOPV reassortant virus developed as a live LASV vaccine candidate.  The manuscript by Johnson et al. characterized ML29 infection of STAT1-/- mice and the effects of ML29 interfering particles (IPs) on viral attenuation and immune responses. The authors have shown that ML29 caused more attenuated infection and stronger LASV-specific T cell responses in stat1 KO mice than the non-pathogenic MOPV, which provide supportive evidence for the safety profile and immunogenicity of ML29. In addition, the authors have shown that ML29 IPs caused further attenuation and strong immune responses in Stat1-/- mice. The studies are novel and comprehensive. Experiments are well designed. Some minor weaknesses are listed below.

1.     Figure resolution is too low to view the details, especially for Figures 1 to 3. It’s difficult to view the morphology of IF foci (line 137) in Fig. 1C. It's hard to read x and y axis and other figure annotations in all figures.

2.     Abstract, line 24, “ML29 was more attenuated…”, is incomplete, please add 
“than MOPV” at the end.

3.     Line 137, which subfigure is“ Figure 1c, F1”?

4.     Lines 137-139, “Differences in the morphology of IF foci between naïve and Vero/ML29 cells after infection and superinfection with ML29, respectively, seems to be related to differences in the expression of NP and GP proteins in different passages of Vero/ML29 (Figure A1F)”. It is unclear which subfigure(s) the statement refers to and how the conclusion is achieved.  Please clarify and provide details on the results and the associated figure # to facilite understanding.

5.     Line 180, ML289 should be ML29.

6.     Lines 244-246 state that “… upregulation of IL-6, TNF-a, and IL-1b at early stages of the infection”. This conclusion does not accurately reflect the data shown in Figure 5, in which TNF-alpha production was actually downregulated for all viruses and at all time points, except for MOPV at day 3, of which the induction was barely 1.5 fold (statistical significance was not provided).  As such the statement on lines 244-246 needs revision.

7.     Throughout the manuscript, viral titer is shown as 1x103, of which 3 should be superscripted to avoid confusion.

8.     FigA2 (D-F), for all three viral infections, significantly higher viral RNAs were detected in kidney (up to 210 fold increase) than in liver or spleen (max 26 fold). In addition, ML29P50 viral RNA was nearly undetectable in liver and in spleen, but was detected at high level (29 fold) in the kidney on day 7. Please explain/discuss the findings.

Author Response

We appreciate all of reviewer 1’s comments and corrections, which have been incorporated in the revised version of the paper. 

1. The figure resolution was degraded when imported in to the Microsoft word document for upload. A High Resolution PDF with images for all figures will be included with the resubmission. This should correct problems with the axis labels being difficult to read as well as problems distinguishing between the IF foci morphology.

2. The suggested addition to the abstract was made.

3. This has been clarified to read “(Figure 1c, Appendix A Figure 1)” so that the two figures it references are easily found.

4. “(Figure A1F)” has been changed to read “(Appendix A Figure 1F)”. This is the supplemental figure.

5. The typo has been corrected.

6. The statement has been changed to accurately reflect the data. Considering the sample size for these is n=3, statistical significance could not be demonstrated here, however, there was  relatively consistent between individuals as demonstrated by the size of the error bars, and is presented as an interesting trend.

7. Superscript text for exponents has been corrected throughout the manuscript.

8. A relative measure of viral RNA was determined using a qRT-PCR cycle plus one method. Individual tissues were extracted and run independently, and are therefore presented as individual sub-panels. In order to address the apparent differences this exposed in liver versus kidney tissues, in situ hybridization with RNAscope was used to visualize viral RNA in these tissues as shown in figure 3. The differences in the qRT-PCR values between liver and kidney do not appear to be a real finding in light of the RNAscope data. The corresponding appendix figure values for qRT-PCR relative virus in tissues can be directly compared to the corresponding tissues in figure 2 as these were assessed concurrently. In all cases, wild type mice had lower relative viral RNA.

Reviewer 2 Report

The authors introduce their work as follows: Lassa virus (LASV), a highly prevalent mammalian arenavirus endemic in West Africa, can cause Lassa fever (LF) which is responsible for thousands of deaths annually.

There is no vaccine nor effective treatment.

The reassortant ML29 is a vaccine candidate.

Persistent infection of Vero cells with ML29 resulted in generation of interfering particles (IPs) Formulation of ML29 with IPs will improve the breadth of the host’s immune responses and further contribute to development of a pan-LASV vaccine

The reviewer suggests the following: The Introduction should be more focused to the work described in the manuscript.

IPs were generated after persistent infection of cells with ML29 – the data demonstrating a similar level of viral load out to 50 passages should be shown.

The manuscript would be strengthened through the detection of the deletions/SNPs that are selected in p50.

A comparison to the immunogenicity (even in in WT mice) post-infection with LASV would strengthen the work.

The supposition that formulation of ML29 with the IPs would make for a rationally designed and better LASV vaccine need to be supported. A LASV challenge would greatly strengthen the case.

In addition the added advantage of formulation of ML29 (and perhaps MOPV) with the IPs produced from ML29P50 should be explored – is the humoral response augmented/altered – is the cellular response increased/?  Etc.

The discussion should be shortened and more focused. As there is no clearly defined correlate of protection post LASV infection a comparison of antibody levels and cellular immune responses does not always translate to similar levels of protection between vaccine candidates, this should be considered and discussed.

Author Response

We appreciate all of reviewer 2’s comments and corrections, which have been incorporated in the revised version of the paper. 

1. In the revised Introduction we made some changes to be more focused to the work described in the manuscript.

2. As we mentioned in the manuscript, infectious titers of IPs were not detectable by plaques assay after passage 15 and till the end of the experiment (passage 50). Work to further characterize the IPs described is ongoing, but we feel that it is outside the scope of this manuscript. The detection of either SNPs or deletions that characterize IPs has been challenging and elusive, however, the real biological attenuation of IP enhanced ML29 preparations is an important discovery that is documented in this manuscript.

3. We agree with this comment, however, this manuscript lays the groundwork that could potentially justify further characterization with LASV which would require considerable resource expenditure with a maximum biocontainment ABSL-4 lab.

4. We agree with the Rev. 2 that LASV challenge experiments is a valuable comment. We feel this study along with our ongoing work to further characterize ML29 IPs is an important first step to this rationally designed vaccine. LASV challenge in a maximum containment ABSL-4 lab would certainly be a future aim of this work after we has demonstrated the nature and potential of these particles. At this time, however, we feel a study with a LASV challenge would be premature.

5. We agree that the added advantage of formulation of ML29 (and perhaps MOPV) with the IPs produced from ML29P50 should be explored. This is an excellent point and a current direction for our research. While outside the scope of the current manuscript, we are investigating these questions.

6. In the revised manuscript we tried to make discussion shorter focusing on contribution of innate immunity, LASV-specific antibody and cellular immune responses and their contribution to protection induced by advanced vaccine candidates. 

Reviewer 3 Report

In this study, Johnson et.al. found that the replication of Lassa virus vaccine candidate ML29 was attenuated in STAT1 knockout mice, which may contribute to development of a pan-LASV vaccine with full coverage meeting the WHO requirements in future. Generally, the data is interesting, the figures are easy to follow and the interpretation of results is proper. While the quality of the manuscript is high, however, there are still some concerns needing to be addressed.

1.     In Fig. 1a, Fig 1b and Fig.2, the font annotations are too small and they are really tough to see. It’s better to plot those figures clearly so that the readers could read them easily.

2.     In Fig. 3, the authors need to include the scalebar for those histology pictures.

3.     In Fig A3, what about the TNF-a and IL-1b expressions in liver?

4.     It’s reported that the Z proteins of pathogenic but not nonpathogenic arenaviruses inhibit RIG-I-like receptor-dependent interferon production (J Virol. 2015;89(5):2944-55). Also, the N-terminal domain of Z proteins from pathogenic LCMV and nonpathogenic PICV could mediate the differential inhibition of macrophage activation (J Virol. 2015;89(24):12513-7). The authors need to discuss them.

5.     There are some spelling mistakes.

a.     In Fig.2 legends, the “DPI, days after infection” should be corrected into “DPI, days post-infection”.

b.     In Fig.2 legends, “1x103 PFU” should be corrected into “1x103 PFU”.  In Fig.4 legends, “4x105 splenocytes” should be corrected into “4x105 splenocytes”. Please also correct similar mistakes in other places.

Author Response

We are thankful for helpful Reviewer 3’s corrections and comments. All these corrections have been addressed in the revised submission.

1. The figure resolution was degraded when imported in to the Microsoft word document for upload. A High Resolution PDF with images for all figures will be included with the resubmission. This should correct problems with the figures being difficult to read.

2. Unfortunately, these images were captured using a camera that did not allow us to include an accurate scale bar. To mitigate this, we included images of tissues from mock infected mice that show the basic tissue morphology is not altered by viral infection. Additionally, for the RNAscope in situ hybridization assay, the relative size of brown spots is less important than the quantity of punctate brown stains, allowing observation of cells containing viral RNA.

3. Appendix figure A3 presents IL-6 in the liver and spleen as well as TNF-α and IL-1β in the spleen. We chose not to quantify TNF-α and IL-1β in the wild type liver tissue due to the low level of viral RNA, lack of detectable infectious virus from the liver, and the relatively small effect of the fold change of these cytokines from spleen of these animals, where we would expect to see the largest differences.

4. The Z protein-induced RIG-I-mediated IFN-I inhibition and down-regulation of macrophages are included in the revised Discussion section (lines 328-331).

5. The typos and superscripts for exponents have been corrected.